# Attitudes toward School Violence against LGBTQIA+. A Qualitative Study

**DOI:** 10.3390/ijerph182111389

**Published:** 2021-10-29

**Authors:** David Pina, María Catalina Marín-Talón, Reyes López-López, Ainhoa Martínez-Sánchez, Lucía Simina Cormos, José Antonio Ruiz-Hernández, Begoña Abecia, Begoña Martínez-Jarreta

**Affiliations:** 1Department of Socio-Health Sciences, Faculty of Medicine, Espinardo Campus, University of Murcia, 30100 Murcia, Spain; david.pina@um.es; 2Applied Psychology Service, Espinardo Campus, University of Murcia, 30100 Murcia, Spain; reyeslopezlopez99@gmail.com (R.L.-L.); ainhoa.martinez2@um.es (A.M.-S.); luciasimina.cormos@um.es (L.S.C.); jaruiz@um.es (J.A.R.-H.); 3Department of Social Psychology and Psychiatry, Faculty of Psychology, Espinardo Campus, University of Murcia, 30100 Murcia, Spain; 4Consolidated Group of Scientific Research on Occupational Medicine (GIIS-063), Aragon Health Research Institute (IIS-Aragon), 50009 Zaragoza, Spain; begoabecia@gmail.com; 5Department of Pathological Anatomy, Forensic and Legal Medicine and Toxicology, University of Zaragoza, 50009 Zaragoza, Spain; mjarreta@unizar.es

**Keywords:** attitudes, qualitative study, school violence, LGBTQIA+, minors

## Abstract

School climate is one of the main concerns in terms of research and intervention worldwide. Although it can be directed toward any student, some groups seem to be more vulnerable, as is the case of the LGBTQIA+ (lesbian, gay, bisexual, transgender, queer and intersex +) students, among others. Attitudes toward violence are a construct of particular importance for action plans focused on improving school coexistence. The aim of this study is to examine attitudes toward school violence against LGBTQIA+ students and their relationship with violent behaviors. For this purpose, 96 Spanish students of Spanish elementary education (PE) and compulsory secondary education (CSE) participated in this qualitative study through focus groups for its subsequent thematic analysis. The results identify four types of attitudes toward violence, such as the use of violence as a form of fun, to feel better, when it is perceived as legitimate, and as a way of relating to the LGBTQIA+ community. In addition, a greater number of negative attitudes and violent behaviors toward homosexual boys and transgender minors are observed.

## 1. Introduction

School violence is a serious worldwide public health problem. Despite the recent increase in interest, this social problem has always been present at schools. Several studies have addressed the prevalence of this type of behavior. Recently, it has estimated that 30% of minors worldwide are exposed to school violence [1]. This number is reduced to 25% in Europe and approximately to 15% in Spain.

School violence is a multifaceted construct that encompasses both teachers and students, with behaviors of victimization, perpetration, physical and/or psychological injuries, among others [2,3]. In addition, Anderson and Huessman [4] point out that, for a behavior to be classified as aggression, the behavior, carried out with the intention of harming, need not cause objectifiable damage. According to the literature, among the behaviors that are included as school violence are humiliation, social exclusion, physical harm, destruction of property, or disruption in the classroom, among others [1].

While it is true that violence in the classroom can be directed toward any student, perpetrators have a greater tendency to bully people belonging to certain minority groups, as is the case of the LGBTQIA+ (lesbian, gay, bisexual, transgender, queer and intersex +) community [5]. This community includes people with emotional, romantic, sexual, or affectional attraction to people of the same or both sexes, as well as people whose gender identity, gender expression, or behavior does not conform to the sex assigned at birth [6]. Thus, a recent systematic review concludes that the victim’s actual or perceived sexual orientation or gender identity is the basis for the violence directed toward these minors [7]. It has even been claimed that children in this group perceive more violence at school. For example, it has been stated that homosexual students experience a greater number of taunts than their heterosexual peers [8].

Studies that have addressed the prevalence of school violence against the LGBTQIA+ community find a greater proportion of victims compared to studies with general population. Specifically, according to data collected in the United States, 74% of minors with a non-normative sexual orientation or identity have been verbally assaulted, 36% physically harassed, and 16% even physically assaulted [9]. In Spain, prevalence rates are between 51% and 58% [10]. In this line, Moyano and Sánchez-Fuentes [7] state that LGBTQIA+ minors are 91% more likely to suffer some type of violence by their peers, and three times more likely to be sexually assaulted, with a higher risk of polyvictimization. These authors have also observed that boys are more likely than girls to be victims, and that verbal harassment is the most reported manifestation in these cases.

Although the consequences of school violence are negative, LGBTQIA+ minors seem to suffer them to a greater extent, probably due to the poor training of teachers in this regard and the previous inexistence of inclusion policies in schools [11]. Among these consequences, poorer educational results and higher absenteeism stand out at the school level, also leading to lower interest in pursuing higher education. In terms of mental health, these minors are more likely to suffer from depression, anxiety, low self-esteem, substance abuse, and/or risky behavior concerning their sexual health [5]. Moreover, LGBTQIA+ youth victims of school violence have even been found to have a significantly higher risk of suicidal ideation and behavior [8,9].

With the aim of improving the well-being of minors, many studies have related school violence to various social, family, school, and personal variables [12,13,14,15,16,17]. Among these variables, attitudes toward violence have been widely reported as a particularly important variable for the improvement of school climate [14,18,19]. Specifically, Orue and Calvete [20] assert that negative attitudes toward LGBTQIA+ minors may explain the relationship between exposure to homophobic behavior at school and perpetration of homophobic bullying. Likewise, UNESCO states that the main attitude underlying violence toward the LGBTQIA+ community is homophobia, understood as the rejection or aversion to people who do not behave according to the established gender roles or who feel sexually attracted to people of the same sex [1].

The relationship between attitudes toward violence and school violence has been previously studied in the literature. Traditionally, studies with quantitative methodologies use self-reported questionnaires. Sometimes, the use of tools with closed questions does not allow delving into the specificity of the attitude-behavior relationship necessary for its study and/or understanding [21]. Qualitative studies allow us to explore attitudes toward violence from the perspective of minors, with a sufficient level of specificity [22,23] and they are a necessary complement to quantitative studies. Nevertheless, no previous studies have been found that assume a qualitative perspective that explore the relationship between attitudes toward violence against the LGBTQIA+ community and the violence against the aforementioned in a school context. In this sense, there are qualitative studies that specifically explore violence toward the LGBTQIA+ community, such as Grossman et al. [24], who, through five focus groups with gay, lesbian, bisexual, and transgender (LGBT) youth, explored experiences related to school violence. This study concluded that the interviewees did not feel part of their school community, lacking the sense of empowerment that stems from belonging to a group of sexual minority youth. In addition, the violent behaviors they reported were mainly insults, hate speech, harassment, and, sometimes, physical violence. This vulnerability that young people felt at school made them distance themselves from school as the main means of coping.

More recently, Juárez-Chávez et al. [25] conducted a qualitative exploration of violence experienced by gay men and transgender women through focus groups and in-depth interviews. The violence experienced in childhood and adolescence was grouped into: violence occurring at home or with family members, school violence, and sexual violence. Considering the violence that occurred at school, it is noteworthy that the participants reported that it was based not only on discrimination for their sexual orientation but also on weight, race, or disability. Furthermore, the impact of gender-specific norms (such as playing certain sports) was identified, as their non-compliance, with the corresponding transgression of stereotypes associated with masculinity, increases the risk of violence. As a consequence, many of the gay men internalized the rejection of behavior considered effeminate, preferring not to maintain contact with other gay men with this type of behavior. On another hand, they indicated differences between violence received by gay men and transgender women, with the latter the object of not only physical violence, such as hitting or pushing, but also of sexual violence, such as groping, with sexual violence being directed almost exclusively at this group.

The main objective of this study is to explore the attitudes and violent behavior toward the LGBTQIA+ community in the school context. Specifically, it aims to identify attitudes toward school violence directed against any member of the LGBTQIA+ community. Taking into account the literature consulted, our main hypothesis is that there will be a direct relationship between positive attitudes toward violence and violent behaviors toward the LGBTQIA+ community. Specifically, we hypothesize that both transgender minors and homosexual boys will be subject to positive attitudes toward violence and to violent behavior of greater intensity compared to other members of the LGBTQIA+ community.

## 2. Materials and Methods

### 2.1. Theorical Paradigm and Study Design

In the present study, we used a qualitative research design for the analysis of attitudes toward violence targeting the LGBTQIA+ community at school. A large part of the published studies about the attitude–behavior relationship uses a quantitative approach [23,26,27]. Quantitative methodology, although essential for the investigation of this phenomenon, hardly captures the participants’ subjective perspective. This is why qualitative studies acquire special interest, as they facilitate the exploration of the minors’ point of view and how they communicate it, leading to a greater depth in the exploration of the meaning and understanding of the information provided. Unlike quantitative studies, the qualitative approach lacks an explicit theoretical, philosophical, epistemological, or ontological framework to guide the obtaining of results [28,29], positioning itself as a fundamental approach to the study of constructs that have been little addressed in the literature.

For the development of the qualitative methodology, the grounded theory perspective was used in the constructivist approach [30], by conducting focus groups [31]. This approach allows exploring specific situations, avoiding the classical problems of the use of self-reported instruments, especially in minors, when evaluating certain constructs [28,32].

The professionals responsible for the research have multiple publications on school violence. In particular, the research team was composed of a clinical psychologist expert in qualitative studies (first author), three psychologists with experience in working with minors (second, third, and fourth authors), and two social psychologists with experience in research on attitudes toward violence (fifth and sixth authors).

### 2.2. Particpants Recruitment

The present research study was conducted in Spain, 4 incidentally selected schools with a general curriculum participated, 2 schools of primary education (PE) and 2 schools of compulsory secondary education (CSE). The mean number of students was M = 663.33 (SD = 458.29). The participants of the study were selected from among all students enrolled in the 4th, 5th, and 6th grade of PE (9–13 years old) and 3rd and 4th grade of CSE (13–16 years old). Following the interests of the study, participants’ previous involvement in incidents of school violence against the LGBTQIA+ community, either as victims or aggressors, was not a requirement. Of the minors invited to assist in the research, only 10 refused to participate.

The final sample consisted of 96 participants, 53.1% male, with a mean age of 11.35 (SD = 2.09, range between 9 and 16 years). More information on the sample is available in Pina et al. (2021).

### 2.3. Procedure

The study has been written following the COREQ guidelines [33] and has been approved by the Research Ethics Committee (ID: 2317/2019). The selection of schools was incidental. All the students, teachers, and parents/guardians of the classes included in this study were provided with written information on the objectives of the study together with an informed consent form. This consent requested the acceptance of both the audio recordings of the focus groups, as well as the use of these results for the scientific publications. The focus groups were conducted during school hours. For their creation, a random selection was made, with a maximum of four participants from each classroom. Students from different classrooms but all from the same grade shared their experiences in the same group. A total of 12 focus groups were conducted, with an average of 8 participants.

The inclusion criteria for these groups were: (a) to be a student at one of the participating centers, (b) to be over 9 years old or under 16, and (c) to have a good level of expression and comprehension of Spanish. The assumed exclusion criteria were: (a) being a minor enrolled in any grade other than those selected (from 4th grade of PE to 4th grade of CSE); (b) having some type of limitation in the comprehension or expression of language (cognitive or physical); (c) rejecting participation or not submit the informed consent form signed by the minor and their parents/guardians and/or (d) missing class on the day the study is conducted.

To carry out the focus groups, each school prepared spaces reserved exclusively for the development of the focus groups. The participants were accompanied from their classroom to this space by a member of the school management team. At the time of the interviews, only the minors and interviewers were present. Before starting the audio recording, they were reminded of the importance of their collaboration, sincerity, and respect for the opinions of their peers, as well as the anonymity of their participation. Emphasis was placed on the anonymity of the responses, reminding them that the recordings would be destroyed after their transcription, eliminating from this document any data that could identify them.

### 2.4. Data Collection

Focus group discussions were conducted to obtain data following the recommendations of Krueger [31]. This type of methodology has been frequently used in the literature [34,35,36,37]. 

In our study, a script was created prior to the formation of the focus groups, in which statements from the previous review of the literature on attitudes toward violence with the LGBTQIA+ group were collected. Following the methodology proposed by Morales [38], the authors prepared conceptual maps and interviews with key informants to complement and adapt the script to the reality and language used by the minors at schools. The script was tested with a pilot group of children other than those included in the present research.

At least two of the authors conducted each of the focus groups. The person who conducted the groups was male, with extensive training and experience with focus groups as shown in different published studies using this methodology. At the beginning of the interaction, a few minutes were spent to create a good atmosphere by asking questions unrelated to the object of study. The interviewer maintained neutrality by trying to remain free of biases and to not interfere in the children’s discourse. The rest of the researchers supported the different focus groups, taking notes and assisting when necessary, during the interviews. Throughout the performance of the focus group, participants were encouraged to provide information, about any circumstances of which they were aware while avoiding singling out or identifying any specific peer.

### 2.5. Interview Content

Previous studies [39] have provided the structure of the script developed, which covers several variables related to attitudes toward school violence. These variables are based on the model presented by [23,40]. All participants were asked an initial question that sought to explore attitudes toward violence when it is directed toward a member of the LGBTQIA+ community (Table 1). This question was complemented by other questions that were intended to serve as facilitators of the participants’ narratives.

### 2.6. Data Analysis

The data analysis followed the thematic analysis proposal of Braun and Clarke [41] with an inductive and constructionist approach. After the focus groups had been conducted, transcriptions of the recordings were made, which served as a first contact with the data collected. These transcripts were made by at least two of the authors of this study, participating in the writing process and the supervision of the transcripts. Data analysis was performed using inductive techniques, generating initial codes to be discussed later by the authors. If there was no consensus, multiple coding was performed. Once the codes were generated, they were grouped into topics and subtopics. In addition, visual aids were used to support of information. For this purpose, a constructivist perspective was adopted, avoiding a mere description of the data by exploring the latent themes of the information collected. This perspective was assumed given the implicit nature of attitudes, hence admitting in the analysis assumptions, structures, and/or broader meanings that support what is articulated in the data.

Once the previous points had been made, the process went back to refine the codes, adjusting them where necessary and ensuring congruence with the data. Generally, the proposed themes in this study were present in most of the focus groups, with the exception of the information that was considered by most of the authors as very important. After this filtering, the information was structured by means of a conceptual map.

Finally, the various topics were named and defined, the saturation of the data was discussed, and a report was prepared. With the completion of all these phases, the topics were associated with different extracts of information to facilitate their subsequent description. This whole process requires moving back and forward over the data to reduce biases and ensure the accuracy of the information collected [41]. For the generation of codes, the NVivo software was used. The participants of the study did not receive the transcripts for review, nor did they participate in the conclusions of the study.

## 3. Results

After the application of the methodology described above, a set of four interrelated blocks of attitudes toward violence against the LGBTQIA+ community. Under this perspective, the attitudes toward the use of violence against the LGBTQIA+ community that we extracted from the thematic analysis are: (a) as a form of fun, (b) to feel better, (c) when it is perceived as legitimate, (d) as a way of relating. See Figure 1.

### 3.1. Attitudes toward the Use of Violence as a Form of Fun against the LGBTQIA+ Community

The first of the extracted topics refers to attitudes toward violence as an instrument through which minors can perceive themselves or be perceived as funny, without this violence necessarily being associated with an intention to hurt or harm others.

These negative attitudes are associated with situations in which there is a demonstration of affection between minors of the same sex, regardless of whether this affection takes place in a context of friendship. As mentioned earlier, these attitudes appear with greater intensity if the demonstration of affection is between two boys, whereas between girls it seems to be more socially accepted, and girls can express their friendship through various affectionate gestures without leading to a manifestation of violent behavior. The behaviors most associated closely with this attitude are teasing, jokes, or insults from other classmates who label them as homosexuals, using this name-calling as a form of aggression, as can be seen in the following example:-*In our class, there are four or five of us that jokingly blow kisses at each other and they call us faggots, we do it as a joke and they say it to annoy us.*

These attitudes are magnified when it is suspected that this display of affection is because two minors of the same sex are actually a couple. In this situation, violent behaviors of greater magnitude appear, such as ridicule, spreading rumors, or discrediting the image, all of which are mainly of a relational nature:-*They started picking on them because they thought they were a couple.*-*Everything started with a sixth grader… who ended up telling someone in our class, and that person in the class told another person in our class, and they told one of them about it, and he didn’t like that.*

Associated with this attitude, a very specific use of language has been identified, characterized by a large number of verbal offenses, using derogatory words to refer to homosexuality, to give it a negative connotation. For this purpose, minors use words such as “faggot, sissy, or lesbian”. It should be emphasized that, according to analyzed data, it is more common to use terms referring to the sexual condition in a pejorative way toward boys than to insult a girl in the same way. In this sense, when girls receive similar aggressive behaviors, they do not seem to be associated with the type of relationship they have with other girls but for some other reason.

-
*A girl called a friend of mine lesbian because her parents were separated.*


Finally, and as mentioned before, this type of attitude is not only used to amuse oneself but also to amuse others, as exemplified:+*If a person is homosexual, what do you think they do to them?*-*Well, you can call him faggot.*-*No, you tell him: Hey! You are so limp-wristed (Laughter)*

In this study, four examples were found in which pejorative terms regarding their sexuality (e.g., faggot, sissy, etc.) were used specifically toward children. On the contrary, two examples were found in which reference was made to terms regarding their sexuality, although these were of a descriptive nature (e.g., lesbian). Likewise, there is a greater variety of terms used to refer to homosexual boys.

### 3.2. Attitudes toward the Use of Violence against the LGBTQIA+ Community to Feel Better

The interviewees consider that violence toward the LGBTQIA+ community is also used to feel better about oneself. One of the aims of this behavior would be to try to feel socially attractive, strong, or superior to the peer group. These attitudes are mainly associated with rumor spreading and verbal violence to send a message of superiority, as seen in the conversation between two interviewees:-*There is a girl in our class who is, well, who likes girls more, so someone in our class picks on her because of that.*+*And why do you think they do that?*-*I don’t know, to make her mad at him or to annoy her.*-*To look cool*

### 3.3. Attitudes toward Violence against the LGBTQIA+ Community Perceived as Legitimate

Sometimes, participants identify the manifestation of violence as legitimate or justified, finding in our study a great variety of associated situations.

When a minor differs physically from what is established for their sex, this is perceived as a sufficient reason to tease, ridicule, or perform other actions. This type of behavior is performed especially when the recipients are boys with characteristics associated with the feminine stereotype:-*There was a boy who had really long hair before, so they would say, ‘Oh, you are a girl, whatever, you have very long hair, you look like a girl.’ Like that all the time.*

Moreover, this legitimization of violence seems to increase when the minor possesses characteristics associated with the female sex together with behaviors in line with this, such as, for example:-*He said he was pregnant and so they always criticize him, they say, “If he’s a boy, how can he be pregnant?”, he wears crop tops and everything.*-*He wears makeup, he wears his hair long, … So, they criticize him a lot.*


*Sometimes, the interviewees claim that violent behavior may increase in intensity (belittling or physical aggression) to force the minor to behave in a way that agrees the stereotype associated with their birth sex. This magnification of violent behavior can be seen in the next comment:*
-
*Everyone would be surprised. And also, I know people who would do anything to make him go back to the way he was before, even beat him up.*



The legitimization of violence is not exclusively associated with the behavior of others. In other words, minors may justify violence because of the differences in thinking, such as another minor considering it appropriate for them to be attracted to someone of the same sex. These violent attitudes are manifested through verbal aggressions, isolation, exclusion, or rejection of minors who think differently.

A variant of these attitudes to highlight is when the violence toward a person of the same sex is legitimized because the aggressor considers that the other person feels attracted to them. This situation is mainly based on the rejection of the idea that a homosexual person might be attracted to you.

In our results, this type of attitude was not observed in girls. When girls perceived that a person of the same sex might be attracted to them, they did not seem to be concerned about the idea of being labeled as homosexual as much as not having sufficient resources to handle the situation.

+
*Would you be okay with having a [girl] partner who liked girls?*
-
*Yes, I wouldn’t mind.*
-
*It would feel strange, wouldn’t it? In case she likes you, I don’t know.*
-
*And you don’t know how to turn her down.*
-
*That’s true, how would you turn her down?*
-
*In the same way as you reject boys. I have friends like that and it gives me… the only thing that scares me is that they like me.*
-
*With male friends that can happen too.*
-
*Yes, but a male friend is a guy.*


Depending on the type of relationship, when this happen between boys, there seems to be a legitimization of violent behavior of greater intensity, going from occasional teasing to behaviors of exclusion and isolation, especially in activities with people of the same sex. Associated with this legitimization is the idea that the members of the LGBTQIA+ community do not defend themselves from aggression, as one interviewee commented “sometimes, you even hit them because you know they are not going to defend themselves”.

### 3.4. Attitudes toward Violence against the LGBTQIA+ Community as a Way to Socialize

Violence can also be used as a way to interact with the rest of peers, either facilitating the relationship or hindering it. In this sense, some children do not believe that it is appropriate to relate to members of the LGBTQIA+ community, as they are often perceived as people who are transgressing the rules. This attitude is present both toward homosexuals and transgender people:-*You can’t get a boyfriend because boys have to date girls.*-*Because if he’s a boy, he has to dress like a boy.*

Minors think it is not right to relate to people who are breaking the rules, considering that they must be punished for it. The demonstration of violence as a way of relating is mainly expressed through exclusion, rejection, and verbal violence.

Specifically, in homosexual minors, the participants’ testimonies indicate that they cannot have heterosexual friends in case they misinterpret the type of relationship. This situation occurs both in girls and boys. However, on the one hand, the rejection of a gay boy by boys has a connotation of repulsion, as observed in the example “Because, what do I know? They may feel something about you, how disgusting it is”. On the other hand, girls do not want to be friends with a lesbian girl for fear of her liking them and not knowing how to turn her down. The social nature of this attitude makes it mainly related to violent relational behaviors such as exclusion, rejection, and boycott of other people’s friendships. In general terms, transgender minors and homosexual boys are more rejected and left out than homosexual girls.

Finally, although a simplified analysis of the topics was made, it should be noted that the attitudes described can interact with each other in multiple situations or behaviors. In other words, a minor could be involved in violence toward the LGBTQIA+ community because they believe that violence is legitimate and, at the same time, try to be perceived as funny by the rest of their peers.

## 4. Discussion

This present study is an exploration of attitudes toward violence against the LGBTQIA+ community in a Spanish sample and their relationship with different violent manifestations. Regarding our study hypotheses, from our results, it is shown that attitudes toward violence in this context are related to the use of violence as a form of fun, to feel better about oneself, when violence is perceived as legitimate, and as a way of relating. In addition, it is observed that homosexual boys and transgender minors are subject both to stronger negative attitudes and violent behaviors. No previous studies have been found that specifically explore this issue from a qualitative perspective. Nevertheless, Pina et al. [39] concluded in a study similar to ours that attitudes toward school violence in the general population are related to violence to feel better about oneself, as a form of leisure or fun, perceived as legitimate, when it targets those who are different, when it has no consequences, as a way of resolving conflicts, as a way of socializing, and as a way of attracting peers’ attention. These results are partially in line with ours, sharing four of the topics obtained. However, there are differences both in the type of behavior associated with the attitude and the intensity of these behaviors depending on the victim.

Along the lines of other studies, the results described herein provide evidence on the relationship attitude-behavior [19,42]. Furthermore, this is especially relevant in the population in which we performed this research, as, according to longitudinal and cross-sectional studies, when children and adolescents consider that face-to-face physical or relational aggression is appropriate, they engage more in these behaviors [43,44]. In this line, previous studies have found a two-way relationship between homophobic attitudes and aggressive behavior toward the LGBTQIA+ community [20]. Orue and Calvete [20] delved into the role of predictor variables of this relationship, such as exposure to homophobic violence at school, exposure to homophobic language at home, and social interaction with people who identify themselves as LGBTQIA+.

The attitude–behavior relationship has proven to be useful for the improvement of school climate and is one of the most effective factors in intervention/prevention programs [14,18]. Hence, our results can facilitate the creation and/or adaptation of these school violence reduction programs to improve the coexistence of LGBTQIA+ minors at school.

Concerning the topics obtained in our analysis, the interviewed minors referred to attitudes toward violence to feel better about themselves. This relationship between violence and self-esteem has been studied on multiple occasions. There seems to be some consensus in the literature about a deficit of self-esteem in minors involved in school violence [45,46]. For example, some cyber aggressors claim to have victimized their targets to feel better about themselves [47,48]. Beynon [49] suggested that minors felt the need to feel better about themselves, although this does not necessarily have to be associated with a deficit in self-esteem. We consider that our results refer to the latter idea; in other words, it does not seem that minors have a negative perception of themselves as much as a need to feel better or superior to the rest. This could happen especially when the minors’ self-esteem is not based on competencies, skills, or qualities in which they stand out.

Regarding attitudes toward violence as a form of fun, it is well-known that, in Western culture, there are countless examples in which, through television, movies, and/or video games, among others, such violence is used as a form of entertainment. Previous studies have indicated this type of attitude as an important predictor of violent behavior in the school context [23]. Generally, these types of attitudes are related to low-intensity violent behaviors such as name-calling or jokes [39]. Nevertheless, when these attitudes are aimed toward the LGBTQIA+ community, they seem to be associated with more emotionally charged behaviors, such as teasing, rumors, or ridicule. The relationship between violence and leisure in this collective has been studied previously. Ballard and Welch [50] contribute that, in the field of massively multiplayer online games, the members of the LGBTQIA+ community perpetrate fewer aggressions. However, they experience significantly higher rates of sex-related cyber-victimization.

Sometimes, beliefs shared by the group may legitimize violent attitudes and behaviors. These normative beliefs can function as a mediator of aggressive behavior [51]. These beliefs may be justified for various reasons, such as a minor being perceived as being different from what is expected of them according to what is socially established [39]. In this sense, although public opinion is increasingly in favor of diversity in terms of sexual orientation, this is not the case for transgender people and their rights [52]. In fact, public attitudes are significantly more negative toward transgender people than toward gays and lesbians [53]. Therefore, the greater the negative attitudes toward LGBTQIA+ people, the greater the legitimization of violence toward them [54]. This idea is reflected in our study, as participants have reported violence toward heterosexual, homosexual, and transgender minors differentially. That is, there is greater legitimization and, hence, violent behaviors of increasing intensity when the victim is homosexual or transgender. Among these groups, greater legitimization is perceived toward transgender minors, followed by homosexual boys, and, finally, homosexual girls. Among heterosexual boys, there is also greater legitimization if they have some characteristic (hair, clothes, etc.) that is more typical of the opposite sex. This type of group belief has been reflected in other qualitative studies with LGBTQIA+ minors [25].

In our results, we have also observed attitudes toward violence as a way of relating. Grossmann et al. [24] stated that school is one of the cultural institutions aimed at socializing and fitting into the community. As a result, in the attempt to achieve this objective, cultural values related to heterosexuality and gender-“appropriate” expression are fomented, rejecting those that do not follow them. Therefore, these authors point out the need for educational politics that ensure that schools promote the inherent value of each student, irrespective of their sexual orientation, gender identity, or gender expression.

The promotion of these values must include the identification of these supposedly legitimate attitudes toward the LGBTQIA+ community along the lines of what is reported in the literature. According to several authors, the legitimization of violence in the school context is the most strongly related factor to the manifestation of violent behavior [23].

### Limitations and Recommendations for the Future

Qualitative studies have multiple limitations. For instance, it is not possible to generalize the results described here, so it would be interesting to replicate similar studies in other countries or social contexts. This type of work would allow us to explore similarities and differences with the results described here. Another limitation is the small sample size. Although qualitative studies are characterized by limited samples, it would be interesting to carry out studies particularly with primary school students or with secondary school students, specifically with larger sample sizes. In our opinion, it would be advisable to complement the results of our study with quantitative studies, either by applying or creating specific surveys of attitudes toward violence against the LGBTQIA+ community or longitudinal studies that study the described variables over time. In addition, it would be interesting to add focus groups with members of the LGBTQIA+ community as participants to obtain a different and necessary point of view to understand school climate. Furthermore, our study does not seem to consider the possibility of reverse causal effects. Therefore, there is a possibility that violence against LGTBQI+ individuals may come from other students from the LGBTQIA+ community [25] and/or it may stem from reasons other than belonging to the group, for example, the use of illegal drugs, depression, risky behavior, etc. [5]. Future research should take into account this multi-causality to determine more accurately this problem.

## 5. Conclusions

Attitudes toward violence against the LGBTQIA+ community in the school context are related to the use of violence as a form of fun, to feel better about oneself, when violence is perceived as legitimate and as a way of relating. Furthermore, homosexual boys and transgender minors are subject both to stronger negative attitudes and violent behaviors.

The results described here could have broad applicability to socio-community interventions. In terms of research, the qualitative approach to attitudes toward violence against the LGBTQIA+ community is a novel contribution to this field of study. This study provides evidence to the previous quantitative studies, allowing us to explore the specificity, complexity, and variety of attitudes. This study facilitates the understanding of school climate and school violence toward the LGBTQIA+ community.

Regarding socio-community intervention, our results suggest that it is important to include a change of attitude toward violence within the programs to improve coexistence in the academic area. As mentioned earlier, meta-analytic studies suggest that modification of attitudes toward violence is an effective perspective to improve the school climate [14,55,56,57].

## Figures and Tables

**Figure 1 ijerph-18-11389-f001:**
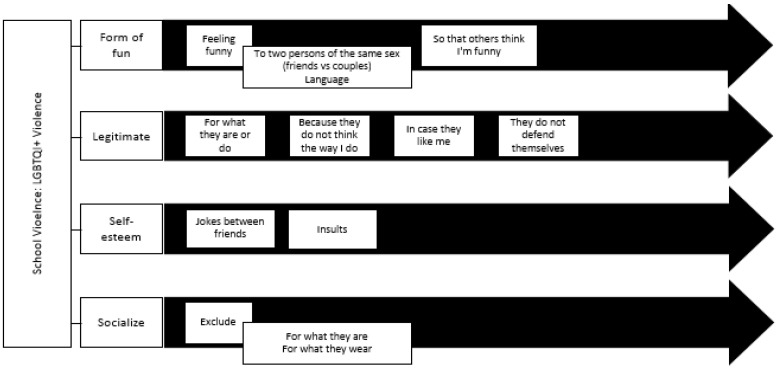
Attitudes toward violence against the LGBTQIA+ community: topics and subtopics.

**Table 1 ijerph-18-11389-t001:** Example of questions asked to the focus groups.

Block	Question
Violence toward the LGBTQIA+ community	Do you think there are problems with LGBTQIA+ students at your school? Why? What happens? When does this happen?
Example of extension question	Do you know of any situation at school where, for someone to feel better than others, something is undertaken or said toward students of this community that makes others feel worse?
Example of extension question	Do you think there is any situation where it is fair or okay to hit, push, insult, etc., another student because of their sexual or gender identity?

## Data Availability

The data presented in this study are available on request from the corresponding author. The data are not publicly available due to the confidentiality agreement with the participants.

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
