# Peer review of "Attitudes toward School Violence against LGBTQIA+. A Qualitative Study"

_ijerph, 2021, doi:10.3390/ijerph182111389_

Round 1

Reviewer 1 Report

The issue which this manuscript focuses on is highly significant and has global resonance. The data that it draws upon appears to rich and offers much scope for analysis and insight. In its current form, though, the analysis lacks depth, most notably in its failure to engage with and build upon the vast body of existing research and literature in this area. Doing so would enable deeper and more interesting insights through recognition and exploration - and contextualisation - of the current data and themes. 

I will address indicative areas of concern in turn:

Issue:

Example:

1. The authors' presentation of material is problematic at times; it is not clear whether this relates to English (language) expression or goes deeper, reflecting conceptual assumptions that are more problematic. The former is significantly easier to resolve than the latter. 

"While it is true that violence in the classroom can be directed toward any student, some groups in particular present a greater vulnerability" (line 43). This sentence inverts the relevant emphasis, so that it is the 'greater vulnerability' of certain groups that is in focus rather than individuals and groups that perpetrate violence. I.e. a variety of victim-blaming  

2. The authors make the claim several times that there are "no previous studies" (line 90) like theirs (i.e qualitative, focusing on young people's perspectives, etc). This is just, frankly, not true. School violence is often considered in the context of bullying rather than in its own right - but it remains the case that there is a substantial body of literature in this area. 

A very quick google search, for example, brings up (top of the page!) -  Logie, C.H., et al. (2016). Exploring Lived Experiences of Violence and Coping Among Lesbian, Gay, Bisexual and Transgender Youth in Kingston, Jamaica, International Journal of Sexual Health, 28(4), 343-353, DOI: 10.1080/19317611.2016.122325

"no previous studies have been found that assume this perspective" (line 90)

NB: Strangely, this is immediately followed by: "There are qualitative studies of violence toward the LGBTQI+ community, such as that of Grossman et al. [etc]" (line 91)

"No previous studies have been found  that specifically explore this issue from a qualitative perspective" (line 403)

3. The authors blur together attitudes to LGBT with attitudes to violence, as if these are one and the same. This is highly problematic, both  conceptually and methodologically. While they state attitudes towards violence as their aim, the questions posed to focus groups (table 1, page 5) largely refer specifically to attitudes towards LGBT students (as justifications for use of violence). If the focus truly was on attitudes towards violence (or school violence), we might expect to see questions about violence in and of itself - i.e. as a way of dealing with situations in general, etc.

"it aims to: (a) identify attitudes toward school violence directed against any member of the LGBTQI+ community ..." (lines 115-116)

"direct relationship between positive attitudes toward violence and violent behaviors toward the LGBTQI+ community" (lines 120-121)

4. The explanation of methodological approach and methods needs some tightening up - in particular, with respect to the highly inaccurate framing of qualitative research  

"Unlike quantitative studies, the qualitative approach lacks an explicit theoretical, philosophical, epistemological, or ontological framework to guide the obtaining of results" (lines 134-137)

5. In the explanation of methods, I am particularly struck by the apparent lack of consideration for ethics and safety concerns. Given that focus groups (i.e. peer groups) were the primary means of data collection, this seems extraordinary. Was no consideration given to the implications of students sharing highly sensitive information in this context? And how might this have influenced the findings? Moreover, how could the anonymity of participants (and their "classmates"), let along the confidentiality, be assured in this environment? Perhaps the authors did attend to these issues, but they haven't made this explicit in the paper. 

"Before starting the audio recording, they were reminded of the importance of their collaboration, sincerity, and respect for the opinions of their peers, as well as the anonymity of their participation. Emphasis was placed on the anonymity of the responses ..." (lines 183-186)

"Throughout the performance of the focus group, participants were encouraged to provide information, both their own and from any classmate they knew" (lines 207-208)

6. Relatedly, there seems to have been little consideration given to diversity within the student group, other than sex and age. Did none of the 96 participants identify as other than male or female (i.e non-binary or gender diverse)?

"The final sample consisted of 96 participants, 53.1% male, with a mean age of 11.35" (lines 158-159) 
7. More generally, some instances of English expression are more substantial than others - i.e. change the meaning

"School coexistence is one of the main concerns in terms of research and intervention worldwide" (line 16)

i.e. coexistence of what ... ?

"School violence is a multifaceted construct" (line 36)

Author Response

We have received and studied yours comments. Thanks to their contributions, this study has significantly improved. We present the comments received point-by-point and how they have been addressed in the article. 

  1. “The authors' presentation of material is problematic at times; it is not clear whether this relates to English (language) expression or goes deeper, reflecting conceptual assumptions that are more problematic. The former is significantly easier to resolve than the latter. Example: "While it is true that violence in the classroom can be directed toward any student, some groups in particular present a greater vulnerability" (line 43). This sentence inverts the relevant emphasis, so that it is the 'greater vulnerability' of certain groups that is in focus rather than individuals and groups that perpetrate violence. I.e. a variety of victim-blaming”

We thank the reviewer for their input on victim-blaming. We have rewritten the part of the text that has been pointed out and revised the rest of the manuscript trying to avoid this.

  1. “The authors make the claim several times that there are "no previous studies" (line 90) like theirs (i.e qualitative, focusing on young people's perspectives, etc). This is just, frankly, not true. School violence is often considered in the context of bullying rather than in its own right - but it remains the case that there is a substantial body of literature in this area. A very quick google search, for example, brings up (top of the page!) - Logie, C.H., et al. (2016). Exploring Lived Experiences of Violence and Coping Among Lesbian, Gay, Bisexual and Transgender Youth in Kingston, Jamaica, International Journal of Sexual Health, 28(4), 343-353, DOI: 10.1080/19317611.2016.122325 "no previous studies have been found that assume this perspective" (line 90) NB: Strangely, this is immediately followed by: "There are qualitative studies of violence toward the LGBTQI+ community, such as that of Grossman et al. [etc]" (line 91) "No previous studies have been found  that specifically explore this issue from a qualitative perspective" (line 403)”

We appreciate your contribution. Indeed, there are previous works that study the experiences of LGBTQIA+ people and violent behavior. However, we would like to emphasize that the aim of this study is to study attitudes toward violence, not received or perceived violence.  Attitudes are an important variable in reducing school violence, according to several studies cited in the manuscript. When we refer to the paucity of previous studies, we are referring to studies of attitudes toward school violence directed toward the LGBTQIA+ community. For this purpose, we have paraphrased some parts of lines 83 to 93 insisting on the lack of articles focused on attitudes toward violence against this group (and not about the subject matter or experiences of this group).

  1. “The authors blur together attitudes to LGBT with attitudes to violence, as if these are one and the same. This is highly problematic, both conceptually and methodologically. While they state attitudes towards violence as their aim, the questions posed to focus groups (table 1, page 5) largely refer specifically to attitudes towards LGBT students (as justifications for use of violence). If the focus truly was on attitudes towards violence (or school violence), we might expect to see questions about violence in and of itself - i.e. as a way of dealing with situations in general, etc. Example: "it aims to: (a) identify attitudes toward school violence directed against any member of the LGBTQI+ community ..." (lines 115-116); "direct relationship between positive attitudes toward violence and violent behaviors toward the LGBTQI+ community" (lines 120-121)”

Thanks for your comment. In this case, we are studying attitudes with a double degree of specificity, on the one hand, attitudes toward violence and, on the other hand, more specifically, that type of attitudes toward a minority (the LGBTQIA+ community, but just as it could be attitudes toward gender-based violence to predict school violence based on gender, socioeconomic status, etc.). In this sense, according to theoretical proposals (see Fazio, 1990), attitudes have been considered as a predictor of behavior and, the greater the degree of specificity of the attitudes and/or behaviors to be evaluated, the greater the predictive power (and, therefore, the more related). For this reason, questions of this type are posed to study attitudes and violent behavior at this level of specificity.

  1. The explanation of methodological approach and methods needs some tightening up - in particular, with respect to the highly inaccurate framing of qualitative research. Example: "Unlike quantitative studies, the qualitative approach lacks an explicit theoretical, philosophical, epistemological, or ontological framework to guide the obtaining of results" (lines 134-137)

Thank you very much for your comment. We have revised the methodology section trying to follow the COREQ guidelines for qualitative studies. Regarding the qualitative approach, you can find it in the manuscript (lines 138-141 and 222-228): “For the development of the qualitative methodology, the grounded theory perspective was used in the constructivist approach [30], by conducting focus groups [31]”, “Data analysis was performed using inductive techniques, generating initial codes to be discussed later by the authors. If there was no consensus, multiple coding was performed. Once the codes were generated, they were grouped into topics and subtopics. In addition, visual aids were used to support of information. For this purpose, a constructivist perspective was adopted, avoiding a mere description of the data by exploring the latent themes of the information collected”.

  1. “Relatedly, there seems to have been little consideration given to diversity within the student group, other than sex and age. Did none of the 96 participants identify as other than male or female (i.e non-binary or gender diverse)? Example: "The final sample consisted of 96 participants, 53.1% male, with a mean age of 11.35" (lines 158-159)””

We thank the reviewer for their input. No, none of the participating minors identified themselves as other than male or female. This was not explored further as the aim of the study is not to learn about the experiences of minors from a particular group.

  1. “In the explanation of methods, I am particularly struck by the apparent lack of consideration for ethics and safety concerns. Given that focus groups (i.e. peer groups) were the primary means of data collection, this seems extraordinary. Was no consideration given to the implications of students sharing highly sensitive information in this context? And how might this have influenced the findings? Moreover, how could the anonymity of participants (and their "classmates"), let along the confidentiality, be assured in this environment? Perhaps the authors did attend to these issues, but they haven't made this explicit in the paper. Examples: "Before starting the audio recording, they were reminded of the importance of their collaboration, sincerity, and respect for the opinions of their peers, as well as the anonymity of their participation. Emphasis was placed on the anonymity of the responses ..." (lines 183-186) "Throughout the performance of the focus group, participants were encouraged to provide information, both their own and from any classmate they knew" (lines 207-208)”

We are very thankful to the reviewer for taking these aspects into account. We would like to emphasize that the ethical and safety considerations were designed with the approval of the Ethics Committee and were followed in detail. Regarding your questions, peer focus groups (focus groups whose members are peers) are the most appropriate for the exploration of certain constructs, as recommended by most authors (these works are available in the bibliography of the manuscript). Both self-report questionnaire studies and individual and group interviews have a high probability of being biased by the sensitivity of the construct to be assessed. In this sense, it is advisable to use this type of methodology because if the group verbalizes socially undesirable attitudes, it can provoke a facilitating effect. For this purpose, multiple groups from different contexts (schools) should be carried out in order to obtain as much variety of information as possible. Likewise, it is always possible that the most sensitive information could not be observed in our study.

Regarding anonymity and confidentiality, the aim of the study is to study attitudes. These are shared in the culture of the school. This is why the questions were always oriented to general events, without pretending to go into specific cases. While it is true that we cannot guarantee that the minors will keep everything reported in these sessions "secret", we followed a rigorous control to avoid pointing out specific people or any other situation that could be taken out of the session. It is possible that the original text is confusing, we have improved its wording thanks to your comment, so thank you very much.

  1. “More generally, some instances of English expression are more substantial than others - i.e. change the meaning. Example: "School coexistence is one of the main concerns in terms of research and intervention worldwide" (line 16); i.e. coexistence of what ...?; “School violence is a multifaceted construct" (line 36)”

We appreciate your recommendation. We have consulted a specialized translator to revise the entire manuscript to prevent this from happening again.

Reviewer 2 Report

My comments disappeared when I clicked on save and submit later.  The paper is great except that it doesn't seem to consider the possibility of reverse causal effects.  In some cases, bullying against LGBTQIA+ students may come from other similar students (see line 109) and some students may bully others (incidentally LGBTQIA+) because of their use of illegal drugs, depression, risky behaviors (line 69) rather than their sexual orientation. 

Such issues ought to be brought up in the discussion section or for future research.  I still blame school leaders for the continuing existence of so much bullying in so many schools.  Why are we paying some administrators large sums (making more than I have ever made as a full professor at a major university) so they can tolerate  bullying of any of their students?

Author Response

We have received and studied yours comments. Thanks to their contributions, this study has significantly improved. We present the comments received point-by-point and how they have been addressed in the article.

  1. My comments disappeared when I clicked on save and submit later.  The paper is great except that it doesn't seem to consider the possibility of reverse causal effects.  In some cases, bullying against LGBTQIA+ students may come from other similar students (see line 109) and some students may bully others (incidentally LGBTQIA+) because of their use of illegal drugs, depression, risky behaviors (line 69) rather than their sexual orientation. Such issues ought to be brought up in the discussion section or for future research.  I still blame school leaders for the continuing existence of so much bullying in so many schools.  Why are we paying some administrators large sums (making more than I have ever made as a full professor at a major university) so they can tolerate bullying of any of their students?”

We wholeheartedly agree with the reviewer. This perspective has also been added in the section on limitations and future lines of research. We thank them for their contribution to our study.

Reviewer 3 Report

Interesting study.

Lines 114-124 should only discuss the first aim. The second and third aims are assumptions that there are associations and differences. Those would be discussed in the results and discussion.

Lines 279-295. I would like more clarity how much more perjorative phrases are used toward boys. How often was it observed toward girls? Is it like Line 344 where a specific attitude was not observed among girls?

Line 366 should be reworded to discussing relating to peers.

Author Response

We have received and studied yours comments. Thanks to their contributions, this study has significantly improved. We present the comments received point-by-point and how they have been addressed in the article.

  1. Interesting study. “Lines 114-124 should only discuss the first aim. The second and third aims are assumptions that there are associations and differences. Those would be discussed in the results and discussion.”

We consider these comments to be very relevant and this information has been incorporated into the study. Thank you very much.

  1. “Lines 279-295. I would like more clarity how much more perjorative phrases are used toward boys. How often was it observed toward girls? Is it like Line 344 where a specific attitude was not observed among girls?”

This comment is especially appreciated. This aspect has been better clarified for the avoidance of doubt.

  1. “Line 366 should be reworded to discussing relating to peers.”

The title in line 366 has been reworded to match the content of the section. Thanks to your comment we have been made aware of this mistake. Thank you very much.

Round 2

Reviewer 1 Report

no further comment